# Stakeholder Perspectives on the Development and Implementation of a Polypharmacy Management Program in Germany: Results of a Qualitative Study

**DOI:** 10.3390/jpm12071115

**Published:** 2022-07-07

**Authors:** Truc Sophia Dinh, Maria-Sophie Brueckle, Ana Isabel González-González, Julian Witte, Marjan van den Akker, Ferdinand M. Gerlach, Christiane Muth

**Affiliations:** 1Institute of General Practice, Goethe University Frankfurt, Theodor-Stern-Kai 7, 60590 Frankfurt am Main, Germany; brueckle@allgemeinmedizin.uni-frankfurt.de (M.-S.B.); gonzalezgonzalez@allgemeinmedizin.uni-frankfurt.de (A.I.G.-G.); m.vandenakker@allgemeinmedizin.uni-frankfurt.de (M.v.d.A.); gerlach@allgemeinmedizin.uni-frankfurt.de (F.M.G.); christiane.muth@uni-bielefeld.de (C.M.); 2Department for Health Economics and Health Care Management, Universitaetsstrasse 25, 33615 Bielefeld, Germany; julian.witte@vandage.de; 3Vandage GmbH, Detmolder Straße 30, 33604 Bielefeld, Germany; 4Department of Family Medicine, Maastricht University, P.O. Box 616, 6200 MD Maastricht, The Netherlands; 5Department of General Practice and Family Medicine, Medical Faculty OWL, University of Bielefeld, Universitaetsstrasse 25, 33615 Bielefeld, Germany

**Keywords:** medication management, multimorbidity, polypharmacy, qualitative research, stakeholder analysis, structured care program

## Abstract

Structured management programs have been developed for single diseases but rarely for patients with multiple medications. We conducted a qualitative study to investigate the views of stakeholders on the development and implementation of a polypharmacy management program in Germany. Overall, we interviewed ten experts in the fields of health policy and clinical practice. Using content analysis, we identified inclusion criteria for the selection of suitable patients, the individual elements that should make up such a program, healthcare providers and stakeholders that should be involved, and factors that may support or hinder the program’s implementation. All stakeholders were well aware of polypharmacy-related risks and challenges, as well as the urgent need for change. Intervention strategies should address all levels of care and include all concerned patients, caregivers, healthcare providers and stakeholders, and involved parties should agree on a joint approach.

## 1. Introduction

Multimorbidity [1] and subsequent polypharmacy are major health concerns and have been linked to negative outcomes such as hospitalization, worsening health-related quality of life, reduced functionality, adverse drug events, and substantial health care costs [2,3,4]. Healthcare professionals and policymakers therefore attach great importance to the management of chronic diseases [5].

In the last 20 years, nationwide disease management programs (DMPs) have been introduced in Germany. DMPs are structured programs for the management of specific chronic diseases including type 1 and type 2 diabetes, breast cancer, coronary heart disease, asthma and COPD. They aim to improve patient care by selecting a care coordinator (usually general practitioners (GPs)), evidence-based treatments, as well as training for physicians and patients. To date, about 7.2 million patients have participated in one or several of these DMPs [6,7].

DMPs are based on clinical medical guidelines [7]. However, management programs have only been developed for single diseases and are based on the corresponding (single disease) clinical guideline. They are therefore inappropriate for patients with polypharmacy and multimorbidity [8].

The EVITA study (Evidence-based Polypharmacy Program with Implementation in Health Care) investigated, how a comprehensive polypharmacy management program (PMP) could be developed and implemented in Germany. In the multicenter project, we examined systematic reviews (e.g., of multimorbidity and polypharmacy guidelines [9]), performed an evidence-based update and upgrade of the existing German polypharmacy guideline [10], analyzed routine data [11], and qualitatively analyzed stakeholders’ perspectives. The aim of the latter, the results of which are presented here, was to investigate stakeholders’ views on the development and implementation of a structured program for the management of patients with polypharmacy in Germany.

## 2. Materials and Methods

This article follows the Consolidated criteria for Reporting Qualitative research [12] (see Appendix A).

### 2.1. Study Design

A qualitative design was used to investigate stakeholders’ views on the content, facilitators, barriers, and implementation strategies of a PMP. Qualitative expert interviews were considered appropriate for investigating the perspectives of stakeholders. In this context, experts are understood to be persons that have specific experiential knowledge as a result of their position or work. They are often involved in decision-making processes and are consulted in their roles as experts and asked to either provide information, to evaluate specific situations and scenarios, or both [13].

### 2.2. Setting

In Germany, DMPs are initiated by the Federal Joint Committee (G-BA). The G-BA selects chronic conditions that are suitable for a DMP and defines the content, requirements, and the framework of the program. The G-BA includes four umbrella organizations: (1) The National Association of Statutory Health Insurance Physicians, (2) The National Association of Statutory Health Insurance Dentists, (3) The German Hospital Federation, and (4) The Central Federal Association of Health Insurance Funds [14]. Certified patient representatives participate in all G-BA sessions, but are not entitled to vote [15]. Statutory health insurers are legally obliged to offer DMPs, but participation is voluntary for physicians and patients [6,7].

### 2.3. Participant Selection and Recruitment

To be eligible for the study, stakeholders were required to be involved in either the development or implementation of DMPs in clinical practice. They also had to have knowledge or experience of caring for patients with polypharmacy and multimorbidity. Our aim was to interview experts whose specific knowledge would enable them to make structured suggestions that could be understood and applied by others [16]. We therefore invited stakeholders from the G-BA (including patient representatives) and statutory health insurers to participate in our study. Furthermore, GPs and pharmacists were recruited because their specialist knowledge was considered most relevant to our topic of interest. Based on the institute’s network, personal contacts, and snowballing, purposive sampling was used to select participants. We contacted stakeholders via email and informed them about the study. If they were interested, an appointment for an expert interview was made. All participants provided written informed consent prior to the interview.

### 2.4. Data Collection

To investigate stakeholders’ views, the stakeholders were interviewed separately by one experienced female researcher (M.-S.B. or T.S.D.) using a semi-structured interview guide (see Appendix A). Due to the COVID-19 pandemic, interviews were conducted via telephone. The interview guide was developed by three researchers (M.-S.B., T.S.D., and C.M.) based on the “Manual for Conducting Qualitative Research” by Helfferich [17], and it used an exploratory approach to investigate the field of interest and generate hypotheses [16]. The introductory question was: “Please tell me what you think about the current healthcare of patients with multimorbidity and polypharmacy” and concluded with questions on specific healthcare challenges, the identification of target groups for a PMP, components of a possible PMP, the choice of healthcare professionals that should be involved, and questions on the implementation of the program. The interview guide was discussed intensively with an interdisciplinary qualitative research group at the Institute of General Practice at Goethe University Frankfurt and was tested in two pilot-interviews. Interviews were recorded and transcribed verbatim by a professional transcription office. To protect the participants’ anonymity, names and institutions appeared in the transcripts in the form of a code (IP for “interview partner” and an individual number). Field notes were taken after the interviews. Interviews were not repeated and transcripts were not returned to participants for verification.

### 2.5. Data Analysis

Transcripts were imported into MAXQDA-18 and were analyzed by two researchers (M.-S.B. and T.S.D.) using qualitative content analysis according to Mayring [18]. The two researchers coded the first interview using deductive-inductive coding, whereby the coding tree was developed based on the interview guide. The following interviews were coded independently and were regularly discussed by both researchers. Coding trees were merged and discussed until consensus was reached on a final analytical framework. Interviews were conducted in German and a native speaker translated quotes into English.

### 2.6. Researchers’ Characteristics

Researchers’ assumptions and interests may have influenced the analysis and interpretation. T.S.D. (female): health scientist and researcher; M.-S.B. (female): physiotherapist and researcher; A.I.G.G. (female): MD, health manager and researcher; J.W. (male): post-doc health economist and researcher; M.v.d.A. (female): health scientist, researcher, and professor of polypharmacy and health services research; F.M.G. (male) and C.M. (female): MDs, health-scientists, researchers, and professors of general practice.

## 3. Results

In total, ten stakeholders agreed to participate in the qualitative expert interviews: two GPs, one pharmacist, one representative of each of three separate German statutory health insurers, three experts from either one of the umbrella organizations or a department of the G-BA, and one certified patient representative of the G-BA. Of the participants, 60% were male. The interviews lasted 25–64 min (on average: 42 min).

In the following, we present stakeholders’ views on the development and implementation of a PMP in Germany. Six key issues were identified: (1) key challenges in multimorbidity and polypharmacy, (2) identification of the target population, (3) components of the PMP, (4) healthcare providers’ and stakeholders’ roles and tasks, (5) barriers, and (6) facilitators. Figure 1 provides an overview of the key issues.

### 3.1. Key Challenges in Multimorbidity and Polypharmacy

Stakeholders identified a range of challenges in the care of patients with multimorbidity and polypharmacy at a patient, healthcare provider, and system level.

#### 3.1.1. Patient Level

Stakeholders expressed concerns relating to drug safety, the vast number of available drugs, different combinations of drugs, as well as possible harm, adverse drug reactions, adverse events, and negative outcomes:

“I think the awareness that we’re dealing with something that‘s really dangerous, or potentially dangerous—apart from being a possible godsend, it’s just also a dangerous weapon.”(IP3, general practitioner)

In this context, patient communication was considered challenging and time-consuming, especially as it related to de-prescribing, eliciting patient preferences, and discussing risk:

“What I find really difficult—in the guidelines it always says you have to come to a joint decision. […] How are you supposed to communicate that in individual cases? You can’t say to the patient, ‘What would you prefer—a stroke or a gastrointestinal hemorrhage?’”(IP5, Federal Joint Committee)

Other challenges that were mentioned on a patient level were that too much was asked of the patients and that they lacked knowledge of both their diseases and how to apply their medications. From the perspective of the pharmacist, one of the challenges was:

“... a lack of knowledge on behalf of the patients and relatives about the medication, so that the medication was used wrongly, taken wrongly. And that just leads to a loss of efficacy or a significant increase in the amount of side effects in the patients.”(IP10, pharmacist)

Older patients, in particular, were considered to be at high-risk because of, for example, age-related physiological changes or difficulties managing multiple medications.

#### 3.1.2. Healthcare Provider Level

In view of the complexity of caring for multimorbid patients with polypharmacy, the lack of a care coordinator was identified as a major challenge. The patient representative considered it challenging that:

“… many multimorbid patients get lost in the system and have the problem that within the system there is no single person that takes responsibility and maintains an overview of, for example, their medication plans.”(IP9, patient representative)

This problem is exacerbated by a lack of continuity of care, especially when multiple healthcare professionals and disciplines are involved, by the limited time that is available for consultations and communication, and because possibilities for delegation are inadequate. This can lead to information gaps and discontinuation of care, particularly when several sectors are involved:

“It’s just that we have the problem that we have cross-sectoral interruptions in care. We have deficits in the transfer of information between individual providers, like specialist-GP, but also between the different health care sectors. And at the end of the day, that’s what causes the problems.”(IP2, statutory health insurance)

Further challenges concerned a lack of clinical decision support and under- and overuse despite polypharmacy:

“The same problems definitely occur when medications that are actually necessary are left out. The special challenge is to find a structure that enables me to do that responsibly, without causing any harm … without giving too little and without giving too much.”(IP5, Federal Joint Committee)

#### 3.1.3. System-Level

Other challenges mentioned by stakeholders concerned unclear financing and responsibilities, and formal problems resulting from the fact that healthcare services for patients with multimorbidity and polypharmacy often involve multiple sectors and are covered by different social security legislation:

“From a formal perspective, that’s a huge problem because you cross from one sector to another, from one set of laws to another. Nursing is covered by a different set of laws to health insurance, which means it goes beyond the remit of Social Security Statute Book 5 (SGB V), and that leads to enormous formal problems […] the gap between nursing and SGB V. Just because of reimbursement.”(IP5, Federal Joint Committee)

Stakeholders further spoke of competing interests among various participants. A representative from a statutory health insurance explained:

“The challenge is the issue of the quality of medical care in an economic framework. And that is in a group of insured persons that are not the focus of every health insurer’s marketing activities.”(IP2, statutory health insurance)

### 3.2. Identification of the Target Population

Stakeholders provided suggestions on who should be included in the target group for a PMP by recommending possible inclusion criteria.

#### 3.2.1. Polypharmacy

Polypharmacy was the most frequently mentioned criterion, whereby almost all interviewees recommended using a minimum number of chronic medications. For example, one pharmacist suggested including patients with more than three medications, whereas GPs considered a higher number more feasible in clinical practice:

“… long-term medication that includes at least five drugs. I can’t think of any other criteria […] What happens at the moment is that we’re supposed to draw up a (medication) plan with (only) three medications. That’s quite simply not manageable.”(IP3, general practitioner)

#### 3.2.2. Morbidity

Morbidity as a criterion was intensively discussed, as stakeholders from different groups and even within the same group had contrasting views. For example, one GP and one representative from the G-BA suggested including patients that had multiple chronic conditions, whereas another GP and G-BA representative felt that multimorbidity would be an unsuitable inclusion criterion:

“I don’t think relying on a number is appropriate because some chronic diagnoses are associated with no particular limitations and don’t really justify the provision of special management. It then makes better sense to focus on providing something on the basis of the indication, and to give priority to indications that result in limitations.”(IP5, Federal Joint Committee)

In this context, a representative from a statutory health insurer supported the idea of including patients based on the “number of involved healthcare providers” (IP2, statutory health insurance).

#### 3.2.3. Age

Stakeholders provided suggestions for selecting patients based on age. Some considered older patients most suitable for a structured polypharmacy management program and would therefore only include patients of 60 years and older. Others were against age restrictions and supported the idea of including younger and middle-aged patients:

“It doesn’t depend on age. It can happen to a 30-year old […] problems with interaction are the same as with an 80-year old […] So if you had to define a second criterion, then I’d say that certainly from 50 on makes sense. Because all the illnesses that we want to draw out via the DMP really start earlier.”(IP3, general practitioner)

#### 3.2.4. Hospital Admissions

A representative of the Federal Joint Committee suggested using “a past acute stay in hospital because of a chronic illness” (IP5, Federal Joint Committee) as an additional inclusion criterion.

#### 3.2.5. Patient’s Willingness to Change

Stakeholders considered a patient’s willingness and ability to change behavior to be important:

“Well, the readiness to change and communicate has to exist, of course.”(IP8, statutory health insurance)

“… it only makes sense when the patient can influence the course of the illness. And that means that he also has a certain control over his lifestyle and that he is actually prepared to change it.”(IP7, Federal Joint Committee)

### 3.3. Components of the PMP

Stakeholders provided suggestions on the content of a PMP. In general, they wished for a program that consisted of separate evidence-based intervention components that are implementable and feasible in daily practice. Table 1 gives an overview of the components mentioned by stakeholders and includes supporting quotes from the interviews.

### 3.4. Healthcare Providers’ and Stakeholders’ Roles and Tasks

Participants expressed their views on the roles and tasks that should be adopted by healthcare providers and stakeholders in a PMP, and classified them according to the development of the PMP and its implementation in clinical practice. Table 2 and Table 3 provide an overview of potential stakeholders and their designated roles and tasks.

### 3.5. Barriers

Stakeholders expressed their views on what may hinder the development and implementation of a PMP.

#### 3.5.1. Complexity and Heterogeneity

Perceived challenges concerned the complex care and heterogeneity of patients with multimorbidity and polypharmacy. Due to different patterns and combinations, no gold standard exists for healthcare in this patient population. Stakeholders found it difficult to define specific, standardized, and measurable parameters that could serve as quality indicators in a PMP. As a result, some stakeholders did not consider multimorbidity and polypharmacy suitable for existing DMP structures:

“I imagine it would be a bit too abstract if I had a polypharmacy or multimorbidity DMP because they are always organized differently. […] The structure of DMP programs is standardized, which means there has to be a precise description of what medical tasks are to be performed and what parameters are to be measured.”(IP6, statutory health insurance)

“Difficult. Unlike other illnesses, there is no laboratory test result, there is no diagnostic method that would permit clear distinctions to be made. […] I assume it would be impossible for a multimorbidity DMP to involve very many specific recommendations for the physician […] because the different configurations are extremely complex. […] so that you certainly couldn’t provide such specific instructions as you can as in the case of coronary heart disease, diabetes or asthma.”(IP5, Federal Joint Committee)

#### 3.5.2. Selection and Measurement of a Potential Outcome

Another difficulty is the development of a useful outcome measure for such a PMP. On the one hand, representatives from the statutory health insurers pointed out that programs should be cost-efficient and, on the other hand, they doubted if the effect of a PMP could be verified and measured:

“the issue of ‘I’m going to organize an appropriate drug therapy and avoid who knows how many hospital stays’. […] To do that to such an extent that you can prove it in this very, very heterogeneous group of patients—that is going to be a real challenge. […] Purely from a practicability point of view. After all, counting tablets is not going to get you anywhere, right? What you have to reduce are potential drug risks and mutually reinforcing side effects in particular.”(IP2, statutory health insurance)

#### 3.5.3. Identification of the Target Population

Stakeholders foresaw difficulties deciding on concrete inclusion criteria and on who should benefit from such a program:

“But I just have to filter out the patients, for whom I actually see an alternative and where I can do something differently, you see?”(IP2, statutory health insurance)

“Clear demarcation of the target group because registration in a DMP is associated with a load of legal consequences and that has to be regulated as clearly as possible. And that’s when it all starts. It’s not all that easy to answer such questions.”(IP5, Federal Joint Committee)

#### 3.5.4. Inadequate Incentives

Some participants thought a PMP would be likely to fail because of issues surrounding compensation and a lack of incentives encouraging healthcare providers to cooperate with one another. They further assumed that participants would have differing views on adequate reimbursement:

“Well health insurers generally have a different view to GPs on what constitutes adequate compensation, you see?”(IP2, statutory health insurance)

“[It] would be plagued by the same problem as plagues the medication plan, namely by the question of compensation for the whole thing. […] that’s what determines motivation for the participating disciplines because no one is going to take the time for case conferences as part of a structured medication review or when adjusting medications.”(IP9, patient representative)

“… well, particularly for the multimorbid, […] for them our health system is pretty strongly—how shall I say it? Reliant on physicians, isn’t it? They play a really, really key role. And the consequence of that is that involving other professional groups, pharmacists if you like, or a social worker, a nurse or whatever, well that could be much more difficult. That could really be a barrier.”(IP7, Federal Joint Committee)

GPs, in particular, were skeptical about the amount of time that would be required, not least for documentation.

### 3.6. Facilitators

During the interviews, stakeholders shared their views on what would be required and what would facilitate the successful development and implementation of such a program. In general, polypharmacy management was considered to be a very important topic. A representative of one of the statutory health insurers noted that:

“…this whole topic is becoming increasingly important. That means more and more initiatives are being developed and more and more people are working on these things.”(IP2, statutory health insurance)

#### 3.6.1. Expected Positive Effects of a PMP

Participants expected a PMP to have a wide range of positive effects and to lead to an improvement in such patient-relevant outcomes as the number of hospital admissions, health status, quality of life and work ability, and overall, to improvement in the management of multimorbidity in everyday life:

“… and that’s of course helpful for the patients over a longer period of time and is of course of great value because he may well be able to achieve a much more stable and improved state of health.”(IP4, Federal Joint Committee)

Further positive effects that were expected to result from the program included an increase in quality of care, e.g., to a reduction in medication-related risks, adverse effects and complications, and to more patient-centered healthcare. One GP expected the PMP to be time consuming, but equally thought the advantages would outweigh such disadvantages:

“It would facilitate the GP’s work—improved cooperation, better overview, a reduction in drug risks and complications. Better treatment.”(IP1, general practitioner)

A PMP might further improve communication between healthcare providers:

“… I think it will lead to incredible savings in resources—in terms of consultations and time spent at the practice, time spent at the pharmacist, time consulting specialists and at the interfaces between them. Huge amounts of time will be saved because information pathways will be clearer and better structured.”(IP10, pharmacist)

With regard to health economics, a PMP was expected to reduce such healthcare costs as those associated with sick pay, unnecessary treatments, follow-up treatment, and hospitalization.

“When our patients feel well then of course we do too. That always sounds like a contradiction but I don’t think it is—in the sense that a patient whose drug therapy is well balanced and that therefore doesn’t have to go to hospital and doesn’t develop a comorbidity—that is of course a positive factor for the health insurance.”(IP8, statutory health insurance)

#### 3.6.2. Practicability and Feasibility

From participants’ perspectives, a PMP can only be successfully implemented if contextual factors relating to both patients and healthcare providers are adequately taken into consideration during its development. They emphasized that interventions need to be practicable, concrete, and feasible in terms of time and efforts.

“It would have to be precisely defined, implementable and most importantly in terms of the time it would involve, requirements would have to be easy to implement for all of the treating professions.”(IP5, Federal Joint Committee)

“And it would have to be easy and practicable enough for the patient.”(IP8, statutory health insurance)

#### 3.6.3. Digitalization

Stakeholders felt that digitalization and the use of digital tools were indispensable to a PMP. On the one hand, they emphasized the benefits of having digital patient files and digitally and centrally available medication plans. On the other hand, they stressed that treating patients with multimorbidity and polypharmacy requires clinical decision support systems that help when (de-)prescribing medications:

“… but, of course, the integration of tools for analysis as well. They are definitely being developed, and they can help physicians by evaluating all the interactions too.”(IP4, Federal Joint Committee)

“.. A digital tool that includes NNTs, that is ‘number needed to treat”(IP5, Federal Joint Committee)

#### 3.6.4. Legal Framework and Incentives

To support the implementation of the PMP, a legal framework that defined responsibilities and incentives, e.g., for healthcare services or inter-professional cooperation, was considered necessary:

“Well, to put it simply to start with—the whole thing is probably going to require appropriate legislation.”(IP4, Federal Joint Committee)

“… it would have to include legal safeguards. Yes, it would certainly have to be fixed in writing who was permitted to do what. And ultimately, and that is always one of the key questions, who is going to be compensated for what and who, at the end of the day, is responsible for the outcome.”(IP10, pharmacist)

“In my opinion, it’s exclusively a question of the incentive systems. If the fee is acceptable, they will do it; if it’s not, they won’t.”(IP3, general practitioner)

## 4. Discussion

This study provides insights into stakeholders’ views on the development and implementation of a structured program to manage polypharmacy in Germany. We were able to identify criteria for the selection of eligible patients, potential intervention components, healthcare providers, and other stakeholders whose involvement was considered necessary, as well as factors that may support or hinder implementation.

In general, all stakeholders were well aware of the risks of polypharmacy to patients and quality of care, and saw an urgent need for change. Moreover, participants identified a wide range of intervention approaches on multiple levels that would affect patients, caregivers, healthcare providers, and stakeholders involved in the care of patients with multimorbidity and polypharmacy. They further agreed that all participants should take a coordinated approach to providing the complex care that this patient group requires.

However, they also identified numerous challenges that would need to be overcome in the development of a structured polypharmacy management program that is comparable to existing German DMPs. One of the difficulties concerned the target population. Some participants suggested selecting the target group based on a defined minimum number of chronic conditions or medications by, for example, using routine data or data from medical records. An analysis conducted as part of the EVITA study and based on routine data showed that 31.7% of those aged 65–79, and 48.4% of patients aged >79 years, fulfilled the criteria of ≥3 chronic conditions and ≥5 chronic medications [11]. While these numerical inclusion criteria may be easy to implement, the high percentage of potentially eligible patients limits feasibility in clinical practice. Instead, the criteria might serve as a starting point for a basic assessment, as suggested by some stakeholders in our interviews, with the ultimate aim of identifying patients with complex (medication-related) care needs that would particularly benefit from a PMP. A combination of several inclusion criteria may further help in the selection of a target group. For example, by adding the criterion of ≥2 hospital admissions within the last year, it was possible to reduce the target group to 16.4 and 21.5% (age groups: 65–79 and >79 years, respectively) [11]. Other inclusion criteria identified in a case study on European polypharmacy management initiatives included, for example, having complex care needs, admission to an acute geriatric or intermediate care unit, and ≥10 medications [19].

In support of our findings, some authors consider a multidisciplinary approach to be necessary and see an urgent need to both improve communication between participating professions and to involve multidisciplinary teams. However, in this respect they mainly consider cooperation between physicians and pharmacists [19], whereas stakeholders in our study stressed the importance of involving healthcare assistants as well. In Germany, healthcare assistants play an important role in primary care, and are often trained to take on additional tasks and to share responsibilities with the GP [20,21].

In both our study and the study of McIntosh [19], medication reviews were considered the most important intervention in polypharmacy management. Their findings indicate that in European countries outside Germany, pharmacists play a larger role in polypharmacy management and, for example, conduct medication reviews. Our study confirms these findings as our participants considered GPs to be responsible for medication reviews and generally assigned pharmacists a supportive role (e.g., adding OTCs to the medication plan). This was partly explained by a “physician-focused healthcare system” as well as unclear responsibilities and financing.

In line with the results of our interviews, the study by McIntosh [19] considered it necessary to align existing structures and systems in order to better address polypharmacy-related challenges. In this respect, they referred in particular to payment and reimbursement systems, incentives, as well as guidance and policies directed at managing polypharmacy. Moreover, our results indicate that efforts should be made to strengthen patient-centered care and improve continuity of care through collaboration among multiple professions and sectors, and by overcoming formal interface problems. Over the long-term, changes on a micro, meso, and macro level were considered necessary.

One of our study’s strengths is the use of a qualitative approach and the purposive recruitment of experts that are involved in decision-making processes and have experience of either developing or implementing DMPs. This approach meant that we included representatives from different umbrella organizations and departments of the G-BA, statutory health insurance companies, and GP practices. We therefore consider our results to provide a comprehensive picture of all aspects that are relevant to the development and implementation of a PMP, which should help increase its feasibility and acceptance. However, as we recruited participants from the institute’s network and through personal contacts, this sampling method may have been prone to bias. Another limitation relates to the small number of participants in our study. With the exception of GPs, the perspectives of different statutory health insurers, umbrella organizations, and departments of the G-BA were only represented by one stakeholder each. It cannot therefore be assumed that their views are representative of their entire stakeholder group.

Our study’s findings may support the development of a comprehensive polypharmacy management program, whereby literature searches should also be used to define the content of the program. A pilot study should investigate the program’s effectiveness and feasibility, and an accompanying process evaluation used to help understand contextual factors and shape the program to increase its acceptance in routine care.

## Figures and Tables

**Figure 1 jpm-12-01115-f001:**
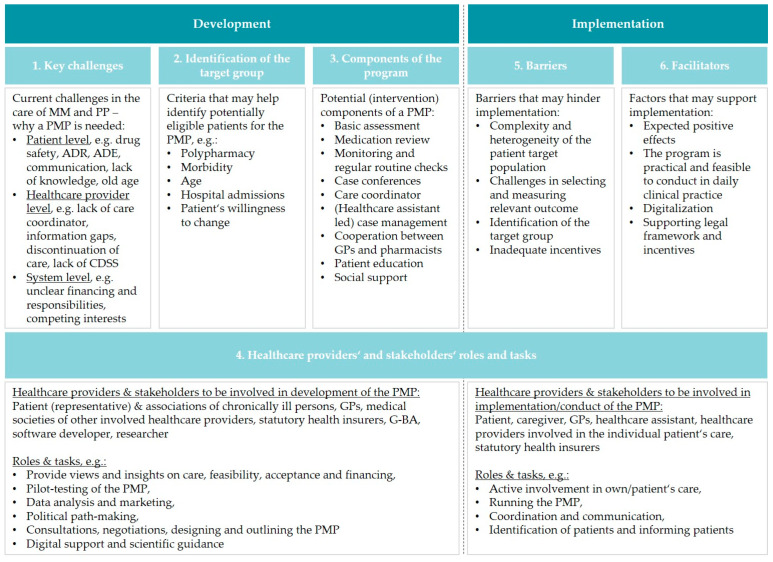
Overview of key issues. Abbreviations: ADE—adverse drug event, ADR—adverse drug reaction, CDSS—clinical decision support system, G-BA—Federal Joint Committee, GP—general practitioner, MM—multimorbidity, PMP—polypharmacy management program, PP—polypharmacy.

**Table 1 jpm-12-01115-t001:** Stakeholders’ suggestions on components of a polypharmacy management program.

Component	Details	Quotes
Basic assessment	Assessment of patient’s individual overall status (e.g., diseases, functionality, quality of life) for the majority of patients, whereby the assessment can/should be conducted in cooperation with other involved healthcare providers	“… in these patients, you really need to have determined the status quo very recently in order to assess which of the underlying illnesses are actually present and which are dominant in the whole affair. You have to find out very precisely how, so-to-say, the patient is feeling, the whole question of functionality, what is his quality of life like, what is he going through? That’s the starting point.” (IP4, G-BA)
Medication review	Overview and inventory of medications, prioritization and (de-)prescribing of medications, checking for interactions and side effects, pharmacological counselling	“… documentation of all the active substances is key. That’s the basic story. And then the weighting: what should be prioritized? And which of the various combinations present risks? Or are there too many active substances, so that the combination of them all is no longer manageable, can’t be evaluated and may possibly lead to problems?” (IP1, GP)
Monitoring and regular routine checks	Regular consultations for monitoring purposes and defined intervals after which medication-related parameters should be checked	“… the most important thing is that the program defines the structure of the intervals: At what intervals does it make sense to routinely check something? […] For what group of medications does it make sense to check which parameters and in what intervals? It would be important to work on something like that for a polypharmacy-DMP.” (IP3, GP)
Case conferences	Case conferences during which healthcare providers work together as a team with clearly defined responsibilities (focus on patient as a whole, not only on medications)	“And you can develop that further in the severe cases—then it’s not a one-man show any longer. In such specialized settings, it might also be possible that several experts are involved, who then work as a team.” (IP4, G-BA) “…such case conferences are ultimately to consider the patient as a whole, and not just necessary for a patient’s medication.” (IP9, PR)
Care coordinator	Named care coordinator (e.g., GP) that has an overview of all of a patient’s medications and healthcare activities, that guides the process, and is the first port of call for patients	“… I think the GP should play a key role in this respect, because in a best-case scenario he is the one that guides, instructs, advises and of course has an overview of the medications and should be competent enough to assess what is feasible and what might be risky […] Yes, in my opinion we’d be making huge progress if the GP were the main contact person as a matter of principle.” (IP1, GP) “Well, the conclusion I draw is that this clientele in particular requires primary care to take on a bigger role.” (IP7, G-BA)
Healthcare assistants-led case management	Management of patients’ care (needs) (e.g., communication with patients, tele-monitoring, home visits) including specific polypharmacy training for healthcare assistants	“But I assume that […] the main instrument to somehow deal with exactly that problem of lack of time and the demand for greater delegation between […] HCAs and physicians […] will definitely be something like case management. That is to say a structure in which non-physician professions proactively approach patients, […] either on the phone or by means of tele-monitoring, or even home visits, and in this structured way gather information from patients, and engage in structured discussions from which it is possible to recognize what the patient sees as the main problem, and what is most important to the patient. And then to reconcile that with the things that the treating physicians nonetheless consider necessary et cetera.” (IP5, G-BA) “Yes, well it’s not only training for patients that we should be thinking about, but also for health care assistants […] on how to interact with patients, how to record information on medications, how to prepare them. That is to say: How to enter them into the medication plan and how to treat it.” (IP3, GP)
Cooperation with pharmacists	Involvement of pharmacists, e.g., for a structured medication inventory including OTC, and regular and occasional consultations between GPs and pharmacists	“The [pharmacists], on the other hand, should provide feedback to the physicians, who know about the therapy and provide therapy in accordance with guidelines, but perhaps don’t have a complete overview of all the things they, the patients, are taking, right? […] Maybe doses should be adjusted. That means there should be close cooperation between physicians and pharmacists.” (IP6, SHI) “Regular consultations between the patient’s GP and regular pharmacist because the GP and the regular pharmacist should be defined, in my opinion. And after every change in the interface (different health care providers) consultations should take place and the polypharmacy and changes should be looked at.” (IP10, PH)
Patient education	Tailored training and independent patient information on the management of medications, empowerment to make shared decisions and to express preferences, and the introduction of patients to self-help groups and patient networks	“… in that case I’d provide the patient with a training program to demonstrate how to deal with polypharmacy.” (IP2, SHI) “And then some kind of training instrument should also be included in this area in my opinion. That means activating the patient so that he participates by doing what he can, so that, when that has been activated, shared and informed decisions can actually be made.” (IP9, PR)
Social support	Support to enable patients with social care needs to manage their everyday lives (e.g., supported by social workers)	“But in addition to physicians, further professional groups should definitely also be involved. And I could imagine someone providing something along the lines of social support. […] Particularly in multimorbid patients, because I believe that in that case, there are many things that aren’t exclusively linked to medical therapies, but like, as I said, things like how to manage your everyday life, you see?” (IP7, G-BA)

Abbreviations: G-BA—Federal Joint Committee, GP—general practitioner, IP—interview partner, OTC—over-the-counter, PH—pharmacist, PR—patient representative, SHI—statutory health insurance.

**Table 2 jpm-12-01115-t002:** Stakeholders that should be involved in the development of a PMP.

Stakeholder	Role/Task
Patient (representative) and associations of chronically ill persons	Express the views of either patients, other chronically ill persons, or both Develop patient information services Provide insights into the reality of care Focus on quality of life and accessibility
General practitioner	Pilot tests the feasibility and acceptance of the PMP Provides expert and experiential input from everyday daily clinical practice to shape the PMP
Medical societies representing other involved healthcare providers	Provide views on feasibility and financing
Statutory health insurers	Data analysis based on routine data to help identify a potential target population Marketing Involved in developing a financial and contractual framework Pilot test the PMP
Federal Joint Committee	Political path-making Following consultations and negotiations, involved in designing and outlining the PMP Implementation into routine care
Software developer	Digital support of the implementation
Research	Scientific guidance/support of the implementation

Abbreviations: PMP—polypharmacy management program.

**Table 3 jpm-12-01115-t003:** Stakeholders that should be involved in implementing/conducting a PMP in clinical practice.

Stakeholder	Role/Task
Patient	Actively involved in her/his own health(care) Participates in patient training Makes informed decisions Prioritizes and communicates preferences
Caregiver	Supports patients with cognitive impairment
General practitioner	Motivates and recruits patients and helps run the PMP Conducts medication reviews (see Table 1) Coordinates patient’s healthcare (incl. gathering information and delegating tasks)
Healthcare assistant	Case management (see Table 1) Communication with patients
Other healthcare providers from the outpatient and inpatient setting	Communication Cooperation Case conferences (see Table 1)
Statutory health insurers	Identify eligible patients for the PMP based on routine data Inform patients and healthcare providers about the PMP Help run the PMP

Abbreviations: PMP—polypharmacy management program.

## Data Availability

Data is not available because we agreed to protect participants’ anonymity and not to share raw data with the public.

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
