# Peer review of "Stakeholder Perspectives on the Development and Implementation of a Polypharmacy Management Program in Germany: Results of a Qualitative Study"

_jpm, 2022, doi:10.3390/jpm12071115_

Round 1
Reviewer 1 Report
The manuscript is well written and referenced. My only criticism is that the authors could show the relationships among the themes uncovered. I believe that MAXQDA19 has this capability. Displaying how the themes interrelate would strengthen their submission.
Author Response
Journal of Personalized Medicine – Manuscript ID 1774555
Title: Stakeholder perspectives on the development and implementation of a polypharmacy management program in Germany: results of a qualitative study.
Thank you very much for the opportunity to revise and re-submit our paper. We would like to thank the editors and reviewers for their thorough appraisals and comments. The feedback was very useful and has enabled us to make important revisions to our paper. For your convenience, we have reproduced the feedback we received in full (queries), and followed it with our responses and any changes made to the manuscript (in blue).
We believe we have responded to all the points raised and have been able to improve the manuscript as a result. If the answers are unclear or insufficient, we are happy to elaborate further.
With kind regards,
Truc Sophia Dinh
(on behalf of all authors)
Reviewer #1
General comment. The manuscript is well written and referenced.
Response. We would like to thank Reviewer 1 for the positive comments and the thorough reading of our manuscript.
Query 1. My only criticism is that the authors could show the relationships among the themes uncovered. I believe that MAXQDA19 has this capability. Displaying how the themes interrelate would strengthen their submission.
Response 1. We have visualized the results from the analyses and have added Figure 1 to the manuscript.
Reviewer 2 Report
An interesting theoretical study about polypharmacy management program with a tile Stakeholder perspectives on the development and implementation of a polypharmacy management program in Germany: results of a qualitative study presented by Dinh et al., in detail which will be informative and surely attracted the attention of readers. However need some minor spell and contextual correction.
Author Response
Journal of Personalized Medicine – Manuscript ID 1774555
Title: Stakeholder perspectives on the development and implementation of a polypharmacy management program in Germany: results of a qualitative study.
Thank you very much for the opportunity to revise and re-submit our paper. We would like to thank the editors and reviewers for their thorough appraisals and comments. The feedback was very useful and has enabled us to make important revisions to our paper. For your convenience, we have reproduced the feedback we received in full (queries), and followed it with our responses and any changes made to the manuscript (in blue).
We believe we have responded to all the points raised and have been able to improve the manuscript as a result. If the answers are unclear or insufficient, we are happy to elaborate further.
With kind regards,
Truc Sophia Dinh
(on behalf of all authors)
Reviewer #2
General comment. An interesting theoretical study about polypharmacy management program with a tile Stakeholder perspectives on the development and implementation of a polypharmacy management program in Germany: results of a qualitative study presented by Dinh et al., in detail which will be informative and surely attracted the attention of readers.
Response. We would like to thank Reviewer 2 for the positive comments and the thorough reading of our manuscript.
Query 1. However need some minor spell and contextual correction.
Response 1. We have visualized the results from the analyses and have added Figure 1 to the manuscript to support contextual understanding. A native speaker has conducted a language review before submission of the manuscript.